# SLAM Modification as an Immune-Modulatory Therapeutic Approach in Cancer

**DOI:** 10.3390/cancers15194808

**Published:** 2023-09-29

**Authors:** Alireza Tojjari, Francis J. Giles, Maysa Vilbert, Anwaar Saeed, Ludimila Cavalcante

**Affiliations:** 1Department of Medicine, Division of Hematology & Oncology, University of Pittsburgh Medical Center (UPMC), Pittsburgh, PA 15213, USA; alirezatojjari@gmail.com (A.T.); maysavilbert@gmail.com (M.V.);; 2Developmental Therapeutics LLC, Chicago, IL 60611, USA; 3Novant Health Cancer Institute, Charlotte, NC 28204, USA

**Keywords:** SLAM, SLAMF, cancer, immune modulation, CLL, AML, MM, HNSCC, HCC, CRC, melanoma

## Abstract

**Simple Summary:**

In the dynamic realm of cancer research, the Signaling Lymphocyte Activation Molecule (SLAM) family has emerged as a significant factor in modulating immune responses within tumors. This review delves into the roles of specific SLAM members, such as SLAMF8 and SLAMF9, and their impact on tumor progression in cancers like colorectal cancer and melanoma. Recent immunotherapy advances show promise, but challenges like resistance persist. Insights suggest that the SLAM family might be a key to overcoming this resistance. Advances in technology, including single-cell RNA sequencing, help to demystify tumor interactions, and SLAM-focused treatments could potentially revolutionize cancer care. This article underscores the importance of patient safety and calls for thorough research to bring the potential of SLAM in cancer treatment to fruition.

**Abstract:**

In the field of oncology, the Signaling Lymphocyte Activation Molecule (SLAM) family is emerging as pivotal in modulating immune responses within tumor environments. The SLAM family comprises nine receptors, mainly found on immune cell surfaces. These receptors play complex roles in the interaction between cancer and the host immune system. Research suggests SLAM’s role in both enhancing and dampening tumor-immune responses, influencing the progression and treatment outcomes of various cancers. As immunotherapy advances, resistance remains an issue. The nuanced roles of the SLAM family might provide answers. With the rise in technologies like single-cell RNA sequencing and advanced imaging, there is potential for precise SLAM-targeted treatments. This review stresses patient safety, the importance of thorough clinical trials, and the potential of SLAM-focused therapies to transform cancer care. In summary, SLAM’s role in oncology signals a new direction for more tailored and adaptable cancer treatments.

## 1. Introduction

### 1.1. Introduction to the SLAM Family

Cellular interactions, instrumental in guiding the immune response, chiefly influence the differentiation and proliferation of immune cells. Such interactions encompass a diverse set of cells, leading to what is termed heterotypic engagements [1]. Central to these interactions is the signaling lymphocytic activation molecule (SLAM) receptor family. The SLAM family includes nine transmembrane receptors labeled SLAMF1 through SLAMF9. Except for SLAMF2 and SLAMF4, which interact reciprocally, these receptors predominantly adorn immune cell surfaces [2]. SLAM1 (CD150) marks activated T cells, B cells, and dendritic cells and is crucial for T-B cell interactions [3]. SLAM2 (CD48) is common among all leukocytes and interfaces with 2B4 on NK cells [4]. SLAM3 (Ly9) is expressed in T and B cells and plays a role in autoimmunity regulation [5]. SLAM4 (CD84) is associated with various hematopoietic cells and platelet aggregation [6], while SLAM5 (CD244, 2B4) is found in NK cells, some CD8+ T cells, and myeloid cells, hinting at its role in activating NK cells [7]. SLAM6 (NTB-A, Ly108) not only triggers NK cells but also modulates autoimmune responses [8]. Additionally, SLAM7 (CRACC, CD319) is predominantly present in plasma cells with ties to conditions like multiple myeloma [9]. Interestingly, SLAM7 seems to exhibit both suppressive and activating roles in the context of cancer [10].

SLAM8 seems vital in modulating inflammatory responses [11], whereas recent studies indicate SLAMF9’s expression in specific cancerous cell lines, suggesting a potential role in melanoma cytokine production and macrophage dynamics [12].

### 1.2. SLAM’s Role in Host Defenses and Immune Responses

The immune system heavily relies on a variety of receptor–ligand pairs to facilitate these cellular interactions. Some of these involve receptors that recognize multiple ligands, representing the phenomenon of heterotypic receptor–ligand engagements. A significant example of this is the T cell receptor (TCR) engaging with antigens presented by the major histocompatibility complex (MHC) molecule [3]. Essential co-stimulatory molecules, like CD28-B7, CTLA-4-B7, and CD40-CD40L, along with their ligands, further enhance this intricate process [13]. Working alongside their intracellular counterparts, especially the SLAM-associated protein (SAP)-related adaptors, SLAM receptors play a pivotal role in modulating immune responses [3]. Their influence is widespread across various immune cells, substantially shaping the outcomes of immune responses in different health and disease contexts. For instance, SLAM6 has been identified as a marker for progenitor exhausted CD8+ T cells, differentiating them from terminally exhausted cells. Through flow cytometry, research confirmed that progenitor exhausted cells, characterized as SLAM6+Tim-3-, coexpress the transcription factor T cell factor 1(Tcf1) and SLAM6, with a significant percentage (76%) of these cells expressing Tcf1. These progenitor cells, when transferred, showed the potential to differentiate into both SLAM6+ and Tim-3+ cells in the tumor microenvironment (TME). Furthermore, upon acquiring the phenotype of terminally exhausted cells, these progenitor cells exhibited increased IFN-γ and granzyme B production, indicating their functional adaptability [14].

The genetic background of patients is of paramount importance in determining the efficacy of therapeutic interventions. Within this context, the SLAM family plays a crucial role in the innate immune response, particularly concerning NKT cell function and macrophage responses [15]. The SLAM family’s genetic variability, mainly manifested through divergent Slam haplotypes, modulates in vivo responses to stimuli like LPS, influencing NKT cell numbers and functionality [15]. Consequently, this impacts macrophage responses to ligands such as the TLR4 ligand LPS [16].

Two exemplary Slam haplotypes, C57BL/6J and 129X1/SvJ, highlight the diversity within the SLAM family. They lead to the differential expression of Slamf6 isoforms, where certain isoforms signal with varying efficiencies [17]. Specifically, the Slam haplotype-2+ found in 129 mice strains is associated with diminished cytokine production, affecting levels of pivotal cytokines like IFN-g, IL-4, and TNF [18].

However, the diverse nature of SLAM haplotypes extends beyond modulating immune responses. Specific haplotypes can lead to varied therapeutic outcomes due to their unique genetic compositions. Notably, certain diseases, such as malaria, have exhibited reduced drug susceptibility in specific haplotypes, highlighting the potential challenges posed by genetic variations in the treatment landscape [19].

Given the significance of the SLAM family receptors, this review endeavors to offer an exhaustive overview of our current understanding of these molecules, elucidating their signaling pathways and influence on immune regulation. Special attention will be paid to their roles in T-cell activation, B-cell differentiation, and natural killer cell cytotoxicity. In addition, we will shed light on the therapeutic implications of SLAM signaling, emphasizing their potential as pivotal targets in a spectrum of diseases, including immunodeficiencies and cancer.

## 2. SLAM Receptors in Oncology: Dual Roles and Therapeutic Potential

SLAM family receptors, traditionally recognized for their immune-modulation roles, have garnered attention in oncology for their multifaceted implications. They have been identified as both potential therapeutic agents and markers for disease prognosis. A growing body of literature underscores SLAM’s ambivalent nature in cancer therapy, portraying them simultaneously as friends and foes [20].

Specifically, the presence of SLAM receptors on antigen-presenting B cells and B lymphoma cells is instrumental for antigen priming and the oversight mechanism of immunosurveillance by CD8 T cells [21]. One noteworthy SLAM family member, CD244, holds paramount importance across various diseases, especially cancers. Its function has been shown to influence disease onset and progression, hinting at its potential utility as a prognostic marker or therapeutic intervention point [22].

Furthermore, a cutting-edge approach has harnessed oncolytic recombinant measles viruses, which, when stripped of their SLAM-binding ability, can hone in on cancer cells expressing particular receptors, notably nectin-4 [23]. Another compelling avenue is the relationship between SLAM receptors and the trajectory of Multiple Myeloma (MM). Unraveling this relationship might be key to innovative immunotherapy strategies, especially targeting disease relapse [24]. 

In light of these findings, it becomes evident that SLAM and its associated family members command a pivotal position in oncology research. Given their roles in immunological modulation, tumor progression dynamics, and potential as therapeutic targets, they form a nucleus in current cancer research narratives. In the ensuing sections, we delve deeper into various cancer types, dissecting specific SLAM receptors’ unique contributions and roles, be it for prognosis or therapy.

### 2.1. Chronic Lymphocytic Leukemia (CLL)

Chronic Lymphocytic Leukemia, predominantly observed in European and North American adult populations, manifests as a marked proliferation of mature B-lymphocytes within the bone marrow, lymph nodes, and bloodstream [25]. The SLAM receptors have gained prominence within this milieu due to their pivotal role in modulating the immune system, especially in the context of CLL.

One such notable receptor, SLAMF1, serves as a suppressor of IL-10 expression and release on the B-CLL cell surface. There is a documented correlation between SLAMF1’s presence and improved clinical prognosis in CLL patients [26]. Notably, an increased expression of both SLAMF1 and SLAMF7 emerges as a positive prognostic sign. Such patients, without the diminished activity of these receptors, may display heightened NK cell-induced cytotoxicity, indicating a more robust immunological oversight. On the other hand, diminished SLAMF1 expression is viewed with apprehension, as it may influence the timing of treatment initiation, as well as overall patient survival rates. Delving deeper into the mechanism, there is a hypothesis suggesting that SLAMF1 loss in CLL could perturb genetic routes that control chemotaxis, autophagy, and responses to treatment [27,28].

A significant interplay to consider in CLL’s pathobiology is between SLAMF1 and the CD180 receptor pathway. This dynamic interaction selectively inhibits both Akt and MAPK signaling. Intriguingly, both SLAMF1 and CD180 have been spotlighted as enhancers of CD20 expression. This might insinuate that B-CLL cells with these receptors are potentially more susceptible to immunotherapies targeting CD20 [29]. These findings suggest that the expression patterns of SLAM receptors can serve as potential biomarkers for predicting disease progression and patient outcomes.

In terms of chemotherapeutic susceptibility, although direct associations between SLAMF1 levels and drug sensitivity in B-CLL were elusive, there is documentation indicating that an elevated mRNA expression of the mSLAMF1 isoform correlates with fludarabine sensitivity and cyclophosphamide resistance. Conversely, an upsurge in nSLAMF1 mRNA expression could signify resistance to fludarabine [29]. On a hopeful note for combination treatments, preliminary research posits that synchronizing an anti-SLAMF6 antibody with the drug ibrutinib effectively stymies CLL cell growth, paving the way for potential CLL therapeutic avenues [30]. 

### 2.2. Acute Myeloid Leukemia (AML)

AML is a rapidly progressing cancer of hematopoietic progenitor cells, leading to the unregulated growth of immature myeloid cells. Although survival rates have improved for chronic myeloid and lymphocytic leukemias, the prognosis for adult AML remains relatively unchanged over the years [31,32]. Within the AML landscape, SLAMF2 is recognized as a positive prognostic indicator. However, it is frequently found to be down-regulated in affected patients. This reduction in SLAMF2 levels might be a tactic employed by AML to avoid detection and elimination by Natural Killer (NK) cells. Intriguingly, methylation is one of the regulatory mechanisms affecting SLAMF2 expression. Studies suggest that hypomethylating agents could boost SLAMF2 levels, potentially intensifying NK cell-mediated attack against AML cells in vitro [33]. Delving deeper into the molecular underpinnings of AML, research indicates the role of the fusion gene AML1-ETO. It is postulated that this gene facilitates the immune evasion of AML cells by specifically targeting CD48 (SLAMF2) [33]. Corroborating this, another study revealed the ability of AML cells to dampen the immune response by epigenetically down-regulating CD48 [34].

### 2.3. Multiple Myeloma (MM)

Multiple myeloma (MM) is a hematologic malignancy marked by excessive growth of clonal plasma cells within the bone marrow. This rapid proliferation leads to severe complications such as bone degradation, renal problems, anemia, and hypercalcemia. Annually, an estimated 34,920 individuals in the U.S. and a staggering 588,161 worldwide receive an MM diagnosis [35]. 

SLAMF3 stands out as a consistent marker on MM cells, regardless of the disease’s phase. Attenuating or eradicating SLAMF3 has a dual advantage: it not only restricts MM cell growth but also makes them more susceptible to drug-triggered apoptosis. A surge in serum-soluble SLAMF3 concentrations could act as an indicative marker for MM’s evolution. Interventions targeting SLAMF3 might offer a way to target the therapy-resistant cells that persist in the bone marrow after standard treatments [36]. Precisely, SLAMF3’s engagement in MM cells activates pathways like ERK signaling and certain transcription factors, supporting cellular survival, growth, movement, differentiation, and resistance against apoptosis. The utilization of anti-SLAMF3 antibodies could hold back MM progression [37]. 

Pioneering studies have crafted SLAMF3-based CAR T cells, showcasing remarkable efficacy against various MM cells both in lab settings and animal models [38]. Another member of the same family, SLAMF5, is also prevalent in MM cells. Cells release the macrophage migration inhibitory factor (MIF), stimulating its expression and leading to Myeloid-Derived Suppressor Cells (MDSCs) aggregation. This consequently up-regulates PD-L1 expression in these cells. Targeting SLAMF5 can counteract this MDSCs buildup, resulting in enhanced T cell activation and a reduced tumor presence [34,39].

Elotuzumab, a specialized humanized IgG1 monoclonal antibody aimed at SLAMF7, has gained approval for use in MM patients. It is prescribed alongside lenalidomide and dexamethasone when 1–3 treatment regimens have been ineffective. Alternatively, it is combined with pomalidomide and dexamethasone after two unsuccessful treatments, specifically those involving lenalidomide and a protease inhibitor [40]. Elotuzumab functions as an immune activator, disconnecting MM cells from the bone marrow’s stromal cells and initiating ADCC against the MM cells [9,41]. Significantly, Elotuzumab amplifies the bond between NK cells and MM cells, driving the NK cells to target and destroy MM cells more extensively than through the typical ADCC process [42,43] (Figure 1).

Toxicities from Elotuzumab include infusion reactions experienced by 10% of patients, with symptoms such as fever, chills, and hypertension. A significant 81.4% of patients reported infections, including opportunistic and fungal types. Additionally, 9.1% of patients developed another invasive malignancy, termed Second Primary Malignancies. Hepatotoxicity was evident in 2.5% of patients, indicated by elevated liver enzymes. Elotuzumab can also interfere with the determination of a complete response in certain myeloma patients. Other commonly reported adverse reactions are fatigue, diarrhea, pyrexia, constipation, cough, peripheral neuropathy, nasopharyngitis, respiratory infections, decreased appetite, and pneumonia [40].

A commendable effort by Bristol-Myers Squibb to provide Expanded Access to Elotuzumab for Multiple Myeloma patients in several countries was initiated and intended for patients who were running out of treatment options [44]. An interesting trial from Tulane University School of Medicine sought to combine Elotuzumab with Selinexor and dexamethasone for relapsed refractory multiple myeloma (RRMM) patients resistant to several treatment lines. However, the study was withdrawn due to a lack of funding from the collaborating pharmaceutical company [45]. It is worth noting that Elotuzumab has the potential to induce ADCC even in MM cells that do not respond to treatments like bortezomib. Its binding mechanism with SLAMF7 directly activates NK cells and, concurrently, prevents MM cell attachment to bone marrow stromal cells due to inherent SLAMF7 interactions [46].

### 2.4. Head and Neck Squamous Cell Carcinoma (HNSCC)

HNSCC, the predominant head and neck tumor histology, has a global incidence nearing 890,000 cases each year. It originates from the mucosa of the upper aerodigestive tract, encompassing areas like the oral cavity, nasopharynx, pharynx, larynx, and lip. Despite significant medical advancements, HNSCC’s lethality remains a significant concern [47]. 

One promising therapeutic direction points to SLAM5 (CD244 or 2B4), a signaling lymphocyte activation molecule family member. This molecule is primarily found in hematopoietic cells, including Natural Killer (NK) cells, certain CD8+ αβ T cells, Dendritic Cells (DCs), and Myeloid-Derived Suppressor Cells (MDSCs). Significantly, CD244 interacts with CD48, found in most hematopoietic cells [48,49,50].

Within HNSCC patients and their mouse model counterparts, there is a marked increase in CD244 levels in tumor-infiltrating CD8+ T cells, which also correlates with heightened PD-1 expression. This elevation is seen not just in these T cells but also in DCs and MDSCs within many tumor sites. A strong correlation exists between heightened CD244 and PD-L1 levels, paired with a surge in immune-suppressive agents. Activation of CD244 in a lab setting was found to suppress the release of crucial pro-inflammatory cytokines in human DCs, and intriguingly, CD244-deficient mice demonstrated delayed HNSCC tumor progression [51].

The potential therapeutic benefits of targeting CD244 have been previously described, especially given its association with releasing immune-suppressive agents in DCs and Mo-MDSCs. This hints at a potential weakening of the immune response within the tumor environment by CD244 signaling [42]. Prior research has indicated a connection between elevated CD244 levels in CD8+ T cells and increased PD-1 expression, suggesting CD244’s role in the exhaustion of these T cells in chronic viral infections and cancer [52,53].

With respect to the myeloid tumor compartment, it is theorized that CD244 signaling augments the immune-suppressive characteristics of both DCs and MDSCs. In the case of DCs, CD244 signaling might reduce the initial activation of T cells and the stimulation of NK cells [48]. Meanwhile, in MDSCs, the presence of CD244 aligns with the suppression of specific antigen-driven CD8+ T cells [48].

### 2.5. Hepatocellular Carcinoma (HCC)

Hepatocellular Carcinoma (HCC) is the most prevalent form of primary liver cancer and poses a significant global health challenge due to its high mortality and limited therapeutic options [54]. Ranking third in cancer-related deaths worldwide, the incidence of HCC is anticipated to rise in the upcoming years [48].

SLAMF3 is uniquely underexpressed in human liver cells, suggesting its potential role in HCC development, whereas other SLAMFs have not been implicated. Notably, SLAMF3 expression is evident in 40–65% of healthy human liver cells, but this is markedly reduced in HCC cell lines [55]. Both mRNA and protein levels of SLAMF3 are considerably lower in HCC cells compared to their healthy counterparts. This observation is further supported by tumor specimens from HCC patients, where SLAMF3 was significantly less present in cancerous areas compared to the surrounding non-cancerous regions [56].

SLAMF3 has been identified as having a significant relationship with cell growth, particularly in the context of HCC. Studies have shown that when HCC cells with high SLAMF3 expression are implanted in mice, the growth of the tumor is inhibited [56]. This suggests that rather than promoting tumor growth, high SLAMF3 expression might suppress it. Additionally, liver cells with abundant SLAMF3 have decreased activity in key cellular pathways like MAPK, ERK1/2, JNK, and mTOR, which are typically associated with cell proliferation. This further underscores SLAMF3’s potential role in regulating HCC cell proliferation [55]. Based on these findings, SLAMF3 not only serves as a distinguishing biomarker between healthy and malignant liver cells but also emerges as a potential therapeutic target in HCC treatment.

### 2.6. Colorectal Cancer (CRC)

Colorectal Cancer (CRC), the most common digestive system malignancy, has both prevalence and mortality rates that eclipse many other widespread cancers [57]. Recently, there have been significant advancements in treatment for certain subtypes, particularly with the emergence of immune checkpoint inhibitors (ICIs) such as anti-PD-1 monoclonal antibodies, leading to frequent updates in treatment guidelines [58]. 

Also known as BLAME or CD353, SLAMF8 is a surface protein within the SLAM family. Despite its potential significance, more research is needed on the role of SLAMF8 within the tumor microenvironment [59]. Recent studies pinpoint the predominant expression of SLAMF8 in tumor-associated macrophages (TAMs), which play a pivotal role in fostering an immune-suppressive tumor milieu. These TAMs are instrumental in establishing an immune-suppressive environment around tumors [60]. With various SLAM family proteins known to influence the tumor immune landscape, their potential as immunotherapy targets becomes increasingly evident.

Further research has indicated a correlation between SLAMF8 expression and the presence of CD8-positive T cells in colorectal cancer. This association suggests that SLAMF8, like other SLAM family members, may influence the tumor’s immune milieu [61]. In essence, emerging evidence correlates SLAMF8 expression with malignancy progression, unfavorable outcomes, and the expression of distinct immune checkpoint markers in CRC. While these findings shine a light on its therapeutic potential, further investigations are imperative to confirm SLAMF8’s viability as an immunotherapy target [61].

### 2.7. Melanoma

Originating from malignant melanocytes, melanomas predominantly arise in the skin’s basal layer [62]. However, they can also manifest in unexpected regions, such as the uvea, digestive system, genitourinary tract, and the meninges—the protective layers around the brain and spinal cord [63]. 

A recent study employing a unique anti-hsSLAMF9 antibody explored human melanoma specimens. Findings indicated the following:

The presence of SLAMF9+ tumor-associated macrophages (TAMs) in 73.3% of human melanomas.

Notably, these macrophages were evident in 95.5% of nevi from melanoma patients and 50% from non-melanoma individuals.

SLAMF9 expression was observed in melanocytic cells of 20% of melanomas and a mere 2.3% of nevi from melanoma patients [12]. 

SLAMF9, a recent debutant to the immunoglobulin superfamily of receptors, is expressed on TAMs across both mouse and human melanomas. Its significant role involves modulating pro-inflammatory cytokines’ release and affecting cellular movement [12]. The widespread occurrence of SLAMF9 in both benign and malignant melanocytic formations complicates its direct linkage with malignancy. Focused research is crucial to determine SLAMF9’s reliability as a melanoma marker [12]. 

In summary, SLAMF9 emerges as a promising component of the immunoglobulin-receptor superfamily. Present within various melanocytic growths, its capability to regulate pro-inflammatory responses and influence macrophage movement, even devoid of internal signaling constructs, is intriguing. Delving deeper into its role, potential binding allies and initiated signaling will be pivotal. Fully grasping the role of SLAMF9+ TAMs in melanoma’s development and progression could spotlight this molecule as a crucial therapeutic target in the near future [12]. Table 1 summarizes data from cancers reviewed here and their relation with SLAM.

## 3. Role of SLAM in Overcoming Immunotherapy Resistance

The emergence of immunotherapy heralds a transformative phase in cancer care, harnessing the body’s natural defense systems to combat tumors. However, a significant hurdle persists: numerous patients are either initially unresponsive to this therapeutic approach or develop resistance over time. Thus, understanding the core mechanisms, especially the role of the SLAM family, is of utmost importance [64].

Current research has brought to light the instrumental role of the SLAM family and intertwined pathways in this resistance paradigm. Notably, Kawase et al.’s 2023 research underscored the IFNγ signaling pathway’s criticality in resistance mechanisms against immune checkpoint inhibitors, suggesting that resistance might largely emanate from a dampened MHC-I expression. A case-in-point was a JAK-negative head and neck squamous cell carcinoma patient who retained HLA-I expression and, intriguingly, showcased a positive response to immune checkpoint inhibitors [65].

Delving into the tumor microenvironment (TME) presents further revelations. For instance, Cappellesso et al. spotlighted the bicarbonate transporter SLC4A4 as a potential game-changer in pancreatic cancer therapy. The strategic inhibition of SLC4A4 can mitigate the acidic TME, facilitating a heightened T cell-driven immune response and diminished immunosuppression. Marrying SLC4A4 targeting with immune checkpoint blockade therapies has emerged as a promising tactic to counteract immunotherapy resistance [64].

In a parallel vein, the ATM signaling cascade’s influence in delineating the differentiation of myofibroblastic cancer-associated fibroblasts has come under the spotlight. Manipulating ATM provides a promising avenue to recalibrate the tumor microenvironment, offering a potential strategy against immunotherapy resistance [66]. Simultaneously, the distinctive properties of cancer stem cells (CSCs) have been mooted as pivotal contributors to this resistance. As Gupta et al. elucidated, factors ranging from the unique surface marker profile of CSCs and the array of cytokines they release to the metabolites they produce all play roles in tailoring the immune landscape within the TME [67].

In essence, the intricate interplay of the SLAM family, intertwined pathways, and specific cellular constituents holds the key to deciphering tumor behavior and immune responses. Venturing deeper into these corridors of knowledge might unlock novel strategies to surmount resistance to immunotherapy, heralding a brighter therapeutic horizon for cancer patients.

## 4. Future Perspectives

As our grasp on SLAM and its affiliated pathways in cancer and immunotherapy resistance intensifies, it beckons a new era of therapeutic interventions. As the field progresses, with tumors showcasing stark heterogeneity and the TME revealing layers of complexity, the horizon of personalized medicine—where therapies are tailored to specific SLAM pathways based on individual tumor profiles—seems closer than ever. Moreover, intertwining SLAM-centric treatments with established immunotherapies, such as immune checkpoint inhibitors, could herald a paradigm shift in heightening therapeutic outcomes and countering resistance.

Preclinical and clinical trials exploring these combinations will be of paramount importance. However, as with all therapies, understanding the mechanisms by which tumors might develop resistance to SLAM-targeted therapies remains crucial. This knowledge will be instrumental in developing next-generation drugs and strategies to bypass resistance. While much focus has been on specific cancers, venturing beyond traditionally studied cancers, the implications of SLAM across a wider gamut of malignancies beckon deeper exploration. As we stride ahead, the sanctity of patient safety remains paramount in developing SLAM-targeted interventions. Concurrently, technological marvels like single-cell RNA sequencing and advanced imaging are poised to unravel the intricacies of tumor interactions and SLAM’s role within—the future of SLAM in oncology research sparkles with promise. With persistent research endeavors, interdisciplinary collaboration, and groundbreaking innovations, SLAM-centric therapies could reshape the contours of cancer care, illuminating a hopeful path for numerous patients globally.

## 5. Conclusions

In the intricate tapestry of cancer biology that constantly evolves and adapts, the signaling lymphocyte activation molecule (SLAM) family has emerged as a pivotal thread, weaving complex interactions between the tumor microenvironment (TME) and the host immune system. As delineated in the preceding sections, members of the SLAM family, notably SLAMF8 and SLAMF9, play a dynamic role in tumor-immune modulation, profoundly affecting the progression and therapeutic outcomes of malignancies like colorectal cancer and melanoma [2,12,60,61,62,63].

The recent advancements in immunotherapy have illuminated the vast potential of harnessing the immune system to combat malignancies [59]. Yet, resistance to these therapies remains a significant hurdle. The SLAM family, with its intimate involvement in immune modulation, has been posited as a potential key to unlocking this conundrum. Kawase et al.’s findings emphasize the IFNγ signaling pathway’s importance and its relation to the SLAM family in deciphering resistance mechanisms [64]. Moreover, insights into the tumor microenvironment, as provided by Cappellesso et al., underline the therapeutic potential of targeting entities like the bicarbonate transporter SLC4A4 to bolster the efficacy of immunotherapies [66].

Personalized therapies, once a distant aspiration in the evolving landscape of cancer treatment, now loom as an impending reality. The heterogeneity of tumors, as suggested by our understanding of SLAM-related pathways, necessitates a patient-centric approach that tailors treatments based on individual tumors’ unique molecular and genetic profiles [67].

Technological strides, where tools like single-cell RNA sequencing complement advanced imaging techniques, provide us with the arsenal to delve deeper into these intricacies, ensuring that our therapeutic interventions are both precise and impactful. However, as we charter these promising waters, it is imperative to emphasize the safety and efficacy of potential treatments. The role of rigorous clinical trials and continuous monitoring cannot be overstated, ensuring that while we strive for effectiveness, the well-being of patients remains at the forefront.

In summation, the SLAM family stands at the confluence of immunology and oncology, offering fresh avenues for research and therapy. With a comprehensive understanding driven by collaborative efforts and innovative technologies, SLAM-targeted therapies have the potential to reshape cancer treatment paradigms. Yet, as we stand at the cusp of this promising frontier, there lies ahead a path demanding meticulous research, unwavering commitment, and an ethos grounded in patient welfare. As the journey unfolds, it is paramount that the scientific community remains both tenacious and adaptive, ensuring that the promise of SLAM in cancer therapy translates from potential to reality.

## Figures and Tables

**Figure 1 cancers-15-04808-f001:**
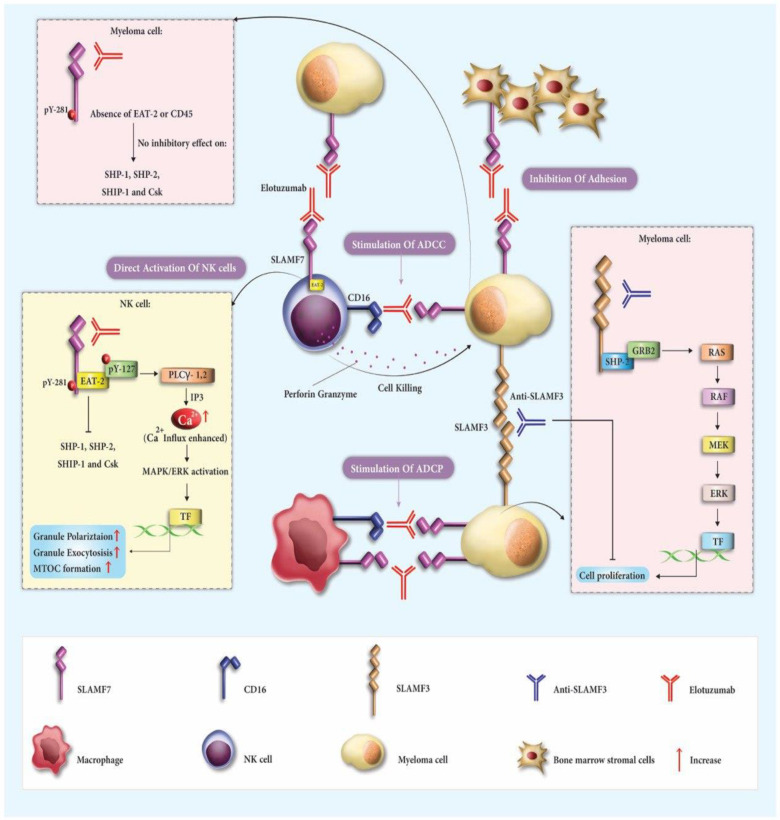
The therapeutic mechanisms involving elotuzumab and Anti-SLAMF3 in multiple myeloma (MM). Elotuzumab engages with various receptors and pathways to coordinate an immune response. It targets signaling lymphocytic activation molecule family receptor (SLAMF7) on both natural killer (NK) cells and MM cells, activating NK cells through EAT-2 and inducing antibody-dependent cellular cytotoxicity (ADCC) via Fc component interaction with CD16. The antibody also disrupts the homotypic interactions between SLAMF7 on MM cells and bone marrow stromal cells (BMSCs), inhibiting adhesion. The signaling cascade initiated by elotuzumab enhances cytotoxicity through the ERK pathway, polarizes cytolytic granules, and enables macrophage-mediated antibody-dependent cellular phagocytosis (ADCP). In the context of MM, SLAMF3 molecules also play a role by forming connections that trigger the ERK signaling pathway and activate distinct transcription factors. The potential for limiting malignancy growth exists through antibodies targeting SLAMF3. Collectively, these mechanisms activate innate immune responses against MM cells and enhance the therapeutic efficacy of both elotuzumab and Anti-SLAMF3.

**Table 1 cancers-15-04808-t001:** A Summary of the role of SLAMF in selected types of cancer (not a comprehensive List).

Cancer Type	SLAMF Involved	Expression Patterns and Notable Findings	Clinical Implications and Outcomes	Therapeutic Strategies and Studies
**CLL**	SLAMF1, SLAMF7	SLAMF1: Differentially expressed in CLL B cells. SLAMF7: Expression correlates with CD38, ZAP-70.	SLAMF1: Potential CLL diagnostic tool. SLAMF7: Correlation with unfavorable prognostic markers.	SLAMF7: Lenalidomide (IMiD) enhances NK-cell-mediated cytotoxicity, targeting the CD20 epitope.
**AML**	SLAMF2	Down-regulated in AML patients.	SLAMF2 down-regulation might aid AML evasion from NK cells.	Hypomethylating drugs might up-regulate SLAMF2, enhancing NK cell-mediated cytotoxicity. AML1-ETO fusion gene targets CD48 (SLAMF2), contributing to AML immune escape.
**MM**	SLAMF3, SLAMF5, SLAMF7	SLAMF3: Ubiquitously expressed in MM cells. SLAMF5: Expressed in MM cells, stimulated by MIF. SLAMF7: Targeted by Elotuzumab.	SLAMF3: Potential prognostic marker for MM progression. SLAMF5: Contributes to MDSCs accumulation and increased PD-L1 expression.	SLAMF3: Potential therapeutic target, especially for therapy-resistant cells. SLAMF3 chimeric antigen receptor (CAR) T cells show efficacy against MM cells. SLAMF5: Inhibition reduces MDSCs accumulation and tumor burden. SLAMF7: Elotuzumab targets SLAMF7, impedes MM cell adhesion, and induces ADCC.
**HNSCC**	SLAMF5 (CD244)	CD244 expression increased in tumor-infiltrating CD8+ T cells, DCs, and MDSCs. Correlates with PD1 expression.	CD244 signaling weakens immune response within the tumor microenvironment, correlating with increased immune suppression and T cell exhaustion.	Inhibition of CD244 signaling might offer therapeutic benefits.
**HCC**	SLAMF3	Expressed in human liver cells but is reduced in HCC cell lines and patient tumor specimens.	SLAMF3 is a marker of healthy liver cells, and its reduction might modulate the proliferation of HCC cells.	Higher SLAMF3 levels were linked with decreased HCC development in lab settings. Inhibition of certain pathways (e.g., MAPK ERK1/2, JNK, and mTOR) in liver cells displaying SLAMF3 hints at therapeutic avenues.
**CRC**	SLAMF8	- Predominant expression in TAMs. - Correlation with CD8-positive T cell presence. - Influence on tumor’s immune milieu.	Emerging evidence correlates SLAMF8 expression with malignancy progression, unfavorable outcomes, and distinct immune checkpoint markers.	Further investigations are needed to confirm SLAMF8’s viability as an immunotherapy target.
**Melanoma**	SLAMF9	- Presence in 73.3% of human melanomas. - Observed in melanocytic cells of 20% melanomas. - SLAMF9+ TAMs evident in a large fraction of melanomas and nevi.	Presence in both benign and malignant melanocytic growths complicates the direct linkage with malignancy.	Deep research is required to determine SLAMF9’s reliability as a melanoma marker and its role as a potential therapeutic target.

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
