# Peer review of "SLAM Modification as an Immune-Modulatory Therapeutic Approach in Cancer"

_cancers, 2023, doi:10.3390/cancers15194808_

Round 1

Reviewer 1 Report

Tojjari et al presented a review manuscript related to SLAM modification in cancer. This is a comprehensive review  focussed majorly on SLAMF8 and 13 SLAMF9, and their implications on colorectal cancer and melanoma.

This review is well-written, well presented and significant to the current research fields. Few minor corrections needed before the final acceptance:-

1. A slight Plagiarism detected in the following are to be rephrased:-

   I. Section 2.3. - Paragraph 4.

  II. Section 3.1. - Paragraph 1

2. Recent references to be cited and included for the year 2023 and 2022.   

Reviewer 2 Report

In the review manuscript entitled “SLAM modification as an immune-modulatory therapeutic approach in cancer”, the authors summarized the roles of SLAM receptors in oncology and stressed patient safety, the importance of thorough clinical trials and the potential of SLAM-focused therapies to transform cancer care. However, there are some publications about the topic. Thus, the novelty is kind of poor. Below are some points which needs to be addressed in order to improve its quality.

1.     The authors might cite the paper (Farhangnia et al, SLAM-family receptors come of age as a potential molecular target in cancer immunotherapy. Frontiers in immunology, 2023).

2.     The title is about SLAM modification in cancer.  However, in the section of introduction (lines 76-78 on page 2), the sentence “emphasizing their potential as pivotal targets in a spectrum of diseases, including autoimmunity, immunodeficiencies and cancer” had some something wrong. They should delete autoimmunity.

3.     Basically, the abbreviation should be defined when the full name appeared for the first time. However, SLAM has been defined twice on page 2 (line 49 and line 80).

4.     SLAM6 has been well known as a marker for progenitor exhausted CD8+ T cells. The authors should describe this point.

Reviewer 3 Report

Dear authors,

The review is well-written and interesting.

However, I have a few major and minor changes to propose to you that, in my opinion, will complete and improve your article.

Major changes:

I suggest modifying the structure of the first section (1. Introduction) in two parts:

  i) an introduction to describe SLAM family

and ii) a section that describes the role of SLMA in host defenses and immune responses, before discussing their roles in oncology in section 2. SLAM Receptors in Oncology: Dual Roles and Therapeutic Potential.

The review lacks crucial pieces of information that, in the current version, are just mentioned. I would develop more the genetic background of patients and, specifically for SLAM family, the existence of different SLAM haplotypes, that should be taken into consideration for further drug development to reduce drug resistance to immunotherapy.

Minor changes:

Figure 1 must be improved as the text is not readable. Please increase the size of the letters.

(lines 312 – 315) Table 1. I would be precise if the list is a comprehensive or not-comprehensive list that summarizes the role of SLAMF in several types of cancer.

Best Regards.

Round 2

Reviewer 2 Report

The manuscript in present form can be accepted.

Reviewer 3 Report

Dear authors,

Thank you for having modified the article as suggested.

It is now suitable for publication.

Best Regards.